# Younger Adults Are More Likely to Increase Fruit and Vegetable Consumption and Decrease Sugar Intake with the Application of Dietary Monitoring

**DOI:** 10.3390/nu13020333

**Published:** 2021-01-23

**Authors:** Louisa Ming Yan Chung, Shirley Siu Ming Fong, Queenie Pui Sze Law

**Affiliations:** 1Department of Health and Physical Education, The Education Unversity of Hong Kong, Tai Po, New Territories, Hong Kong, China; smfong@eduhk.hk; 2School of Nursing, Tung Wah College, Homantin, Kowloon, Hong Kong, China; queenielaw@twc.edu.hk

**Keywords:** younger adults, dietary monitoring apps, behavioral feedback, fruit and vegetable consumption, nutrition knowledge

## Abstract

Establishing healthy eating habits is considered to be a sustainable strategy for health maintenance, and mobile applications (apps) are expected to be highly effective among the young-aged population for healthy eating promotion. The purpose of this study was to investigate the effectiveness of a dietary monitoring app on younger adults’ nutrition knowledge and their dietary habits. A controlled-experimental study was performed with one experimental group having a three-hour nutrition seminar and 12 weeks of dietary monitoring with the app, and one control group receiving a three-hour nutrition seminar. Behavioral feedback delivered by the app was evaluated in facilitating the transfer of nutritional knowledge to nutrition behavior. A total of 305 younger adults aged from 19 to 31 were recruited. Baseline and post-intervention nutrition knowledge and dietary behavior were collected. All mean scores of post-GNKQ-R increased from baseline for both the control and the experimental groups. The mean differences of sugar intake, dietary fiber intake, and vitamin C intake for the experimental group were significantly more than those for the control group (all *p* < 0.001). In addition, the experimental group increased fruit and vegetable consumption significantly more than the control group (all *p* < 0.001). For those younger adults with a relatively large body size, they were more likely to increase fruit consumption with the application of dietary monitoring.

## 1. Introduction

In recent decades, young adults aged 18–24 are found to be gaining weight more rapidly than their prior generations [1]. Indeed, weight gain is prominent among those aged 18–35 in most developed countries. For instance, 35% of U.S. young adults and 27.9% of the Australian young population are shown to be obese [2,3,4]. Moreover, the Population Health Survey [5] revealed that 50% of Hong Kong adults aged 15–84 were either overweight or obese. Particularly in populations under 30-years-old, insufficient intake of wholegrains, fruits, and vegetables, but abundant intake of discretionary foods, such as sugar-sweetened beverages, are ubiquitous [6,7]. In addition, in the transition from attending high school to beginning their employment, young adults also have greater food intake outside of the home [8], inevitably leading to higher energy intake from convenience foods, alcoholic beverages, take-out food, and other high-density, low-nutrient foods [9]. This trend is particularly alarming because a higher body mass index in early adulthood increases the risk of progressive metabolic syndrome by 21 times for men and 27 times for women in the subsequent 15 years [10]. Therefore, establishing healthy eating habits is considered a sustainable and highly effective strategy for health maintenance and the prevention of chronic illnesses [11,12].

In Hong Kong, the government established a committee to create a strategic plan to reduce dietary salt and sugar intake in the Hong Kong population [13]. The work of the committee included public education to promote awareness of healthy eating and transferring knowledge about balanced diets [14]. Indeed, acquisition of nutrition knowledge plays a pivotal role in positively changing people’s eating behavior [15]. Younger adults, however, are found to lack skills in planning, purchasing, preparing, and cooking healthy meals at home, and tend to select nutritionally-inadequate pre-packaged foods [16]. Younger adults also seem to constitute a group that is lacking family support in consuming healthy meals, while simultaneously also not ready to assume responsibility for eating healthfully [17,18]. Self-efficacy constitutes the essential determinant to promote nutrition behavior and education is found effective to increase one’s self-efficacy [19]. Younger working groups have been shown to require innovative nutrition education to make healthy food choices [19]. Recently, nutrition education has radically transformed from descriptive to interactive modalities, i.e., from face-to-face to instant messaging, social media, the Internet, and mobile applications (apps) [20,21,22,23]. Indeed, rapid advancements in technology have markedly shifted health communication in a timely, interactive, and interdisciplinary direction. This integration of knowledge transfer about healthy eating with mobile apps is essential for those technology-immersed younger population [24]. Healthy eating promotion, if conducted with appropriate technology applications, facilitates users to acquire information flexibly in time, pace, and place. Another major advantage of technology application is its specific fit to individual needs, and it allows reflection [22]. To promote healthy eating efficaciously, nutrition knowledge necessitates both education and reflection [25]. Through reflective practices, positive changes in eating habits can be expected [25].

Although mobile apps are anticipated to be suitable for the young-aged population for healthy eating promotion, commercial apps are largely absent in theoretical support and lack sustained engagement with users [26]. Indeed, limited studies exist that target younger adults to change their nutrition knowledge or promote healthy eating with applications of technology-assisted nutrition education. There are also few interventions that apply mobile application (apps) to improve younger adults’ nutrition behavior. In this study, a dietary monitoring mobile app, called the “eDietary Portal”, integrated with a behavioral feedback component, was evaluated in younger adults aged 19–39. The primary aims were to investigate the effectiveness of a dietary monitoring app on younger adults’ nutrition knowledge and dietary habits.

## 2. Research Questions

Does the dietary monitoring app enhance nutritional knowledge among younger adults?Does the dietary monitoring app increase the consumption of fruits and vegetables among younger adults?Does the dietary monitoring app decrease the consumption of salt and sugar among younger adults?Does the dietary monitoring app increase the consumption of whole grains among younger adults?

## 3. Methods

### 3.1. Design and Setting

This was a quasi-experimental two-group design, with one experimental group having nutrition education and the dietary monitoring app as an intervention, and one control group receiving usual nutrition education. The study design aimed to determine whether the behavioral feedback delivered by the app was effective in facilitating the transfer of nutritional knowledge to nutrition behavior.

The sampling was convenience sampling at 12 selected locations with alternate assignment to the control group and experimental group, until the target number for each group (*n* = 150) was reached. Physical measurements and nutrition education were conducted in activity rooms in community centers and lecture rooms in institutions. Ten non-profit organizations (NGOs) and two tertiary institutions were invited to recruit participants and to provide venues for the research team to carry out nutrition education. The NGOs were those providing community services to populations living in the districts.

### 3.2. Participants

A total of 305 younger adults aged 19–31 were recruited as participants. Inclusion criteria were as follows: Hong Kong younger adults aged 18–39; able to read and write Chinese; possess a hand-held device with iOS or Android platform to access the dietary monitoring app; and no prior education on nutrition. Exclusion criteria were as follows: those with special diets due to religion; having one or more chronic disease; and following a prescribed meal plan for weight reduction.

### 3.3. Intervention

Three hours of nutrition education and 12 weeks of self-dietary monitoring practice constituted the two activities implemented for the experimental group. The methodologies of this project followed a series of published works [26,27]. The protocol, including a three-hour nutrition lesson with 12 weeks participants’ continuous reflection of their own diets, was found to be effective to change obese adults and adolescents’ dietary habits and nutrition knowledge [28,29]. The nutrition lessons were arranged in a classroom setting, covering nutritional knowledge regarding food choices, the benefits of common nutrients, diet and diseases, healthy diets, as well as the relationship between diet and exercise. The nutrition lessons transferred the participants’ basic knowledge of healthy eating and methods to maintain a healthy lifestyle with exercise.

Physical measurements of the participants and their age, sex, and daily physical activity levels were input to the app for both the control group and the experimental group. Participants in the experimental group recorded their daily intake on the dietary monitoring app by taking photographs of their foods and inputting the intake servings. Dietary reports with nutrient analysis were shown online for the participants to review. The participants reflected on their dietary intake by reviewing the online reports to check if there was excessive intake of undesirable nutrients or deficient intake of desirable nutrients. They were asked to change their food choices day by day and eat towards their healthy goal. Their self-reflection and change in eating patterns were recorded by the app in a central database. The nutritional knowledge that the participants received in the nutrition education, and reflection in the daily recording and reviewing process, was intended to facilitate the participants to acquire healthy eating behavior. Nutrition behavior established in the process of reflective learning generated by the reflective feedback was expected [30]. Participants in the control group attended the three-hour nutrition education with content that was similar to that in the experimental group. The participants in the control group only recorded their diet in the first week and the twelfth week as baseline and post-intervention measurements, respectively. They did not monitor their dietary pattern between the pre-post measurements. No online report with behavioral feedback was involved.

### 3.4. Outcome Measurements

#### 3.4.1. Nutrition Knowledge

Nutrition knowledge was measured by the General Nutritional Knowledge Questionnaire-Revised (GNKQ-R) [31]. This instrument has been demonstrated to be reliable and accurate to measure nutrition knowledge, and it is also sensitive to changes in nutrition knowledge. The 88-item version used in the current study was a revised version of the original version which comprised 111-items [32]. This revised version reduced the time of self-completion by the younger adults, while still maintaining a consistent, reliable, and sensitive test of nutrition knowledge change [33]. GNKQ-R had high internal reliability with Cronbach’s alpha = 0.93, and that of each section ranged from 0.70 to 0.86. The intraclass correlation coefficients for the external reliability ranged from 0.72 to 0.89 [31]. The construct validity was tested with an effect size ranging from 0.5 to 1.2. [31]. The GNKQ-R comprised four sections. Section 1 consisted of 21 items about younger adults’ knowledge concerning expert advice on food choice. Section 2 consisted of 39 items about younger adults’ knowledge about food group classification. Section 3 consisted of 13 items about choosing healthy foods. Finally, Section 4 consisted of 21 items about knowledge of health problems or diseases related to diet and weight management.

#### 3.4.2. Dietary Behavior

Dietary behavior was measured in collective outcome variables: calorie intake and 11 nutrients intake. Energy requirements were calculated based on participants’ age, sex, physical measurements, and daily physical activity level [34,35]. Calorie intake and nutrient intake reflected the quality of food choices, and these constituted indicators of a healthy diet. The 11 nutrients included carbohydrates, proteins, total fat, saturated fat, trans fatty acid, cholesterol, sugar, dietary fiber, sodium, calcium, and vitamin C. Nutrient analysis was conducted with Nutrition Pro Software and the Food Database of Center for Health Protection. Mean values of daily calorie intake and each of the 11 nutrients were calculated at baseline (week 1) and post-intervention (week 12).

### 3.5. Procedure

The study design and implementation were approved by the Human Ethics Research Committee of the Research Development Office at the serving institution of the first author. Cross-sector collaboration was conducted with non-profit organizations. The health educators were responsible to recruit younger adults in their serving communities. Interested participants received an information sheet describing the objectives and details of the nutrition education program, and signed consent forms before the program commenced. The participants self-administered the questionnaire on general nutritional knowledge as a baseline measurement. The nutrition lessons were taught by the health educators. The participants were introduced to the eDietary Portal for dietary self-monitoring, which was operated on mobile phones or tablet devices. The participants were taught how to use the eDietary Portal to record their food diet on a daily basis. Participants were requested to record their daily diet by capturing and uploading photographs of the foods that the participants ate. They could record their daily diet at home towards the end of each day. The nutrient analysis of the food items was carried out by the researchers.

To enhance healthy individuals to be in compliance with the program, strategies were added to the program to maintain participants’ interest in modifying their eating habits. First, physical measurement and screening could be conducted by participants’ appointment. Body weight, body fat, and blood pressure could be measured every month. For those having abnormal physical measurements, regularly measurements motivated them to continue modifying their eating habits. Second, regular tips about eating topics were sent to all participants by instant messaging. All participants received information on how to eat healthfully on special occasions, different settings, as well as deconstruction of prominent myths about healthy eating. With the target age group and convenient mobile communication, the project team conducted mobile nutrition education, rather than seminar-basis education, to enhance the compliance of recruited participants.

User satisfaction interviews with three participants were carried out. These three interviewees were randomly invited. Because of the COVID-19 pandemic during the time of the user satisfaction interview, all of the interviews were conducted individually with the researcher by telephone. Interview scripts were recorded as audio files for verbatim transcription.

### 3.6. Data Analysis

ANCOVA analysis was carried out between groups on post-GNKQ-R scores as the dependent variables and the pre-GNKQ-R scores as controlled independent variables. ANCOVA analysis was also performed between groups on post-energy requirement, post-energy intake and post-nutrient intakes as dependent variables, and pre-energy requirement, pre-energy intake, and pre-nutrient intakes as controlled independent variables, respectively. Stepwise regression was carried out to elucidate how the independent variables (intervention groups, age, gender, nutrition knowledge, and BMI categories) contributed to significant variances on participants’ post-fruit consumption and post-vegetable consumption. All significance levels were set at α = 0.05.

## 4. Results

In total, 305 younger adults were recruited to participate in this project. The mean (s.d.) age was 22.1 (2.11), and the age range was 19–31. After group allocation, 154 (50.5%) participants were assigned to the control group, and 151 (49.5%) participants were assigned to the experimental group. The mean (s.d.) age of younger adults in the control group was 22.3 (2.34), while that of younger adults in the experimental group was 21.9 (1.84) (*p* > 0.05). Demographic characteristics of the participants are presented in Table 1. Chi-square tests were used to determine if uneven demographic characteristics were distributed between the two groups. The ratio of male to female participants in both the control and the experimental groups was approximately 1:2 (*p* > 0.05). The ratio of single to married was approximately 4:1 in both groups (*p* > 0.05). The majority of the participants were single and had no children, which could be expected because the target group of this project was younger adults aged 18–39, for whom becoming married and having children were comparatively uncommon. The education level for the experimental group was found significantly higher than that of the control group (*p* < 0.05).

### 4.1. Impact of the Dietary Monitoring App on Nutrition Knowledge

All mean scores of post-GNKQ-R increased significantly from baseline for both the control group and the experimental group (Table 3). ANCOVA analysis demonstrated that all scores in knowledge about healthy food recommendations, food group classification, choosing healthy foods, and diet-related health problems or diseases from the experimental group increased to a significantly greater extent than those from the control group (all *p* < 0.001).

### 4.2. Impact of the Dietary Monitoring App on Dietary Behavior

All mean values of energy requirement, energy intake, and nutrients of baseline and post- intervention for both the control group and the experimental group are listed in Table 4. Highlights of nutrient intake changes related to the research questions were sugar, sodium, vitamin C, and dietary fiber were summarized here. Significant decreases in sugar consumption were found in the control group and the experimental group (*p* < 0.001), but the decrease in the experimental group was more significant than that in the control group (*p* < 0.001). In addition, the mean difference of sugar intake for the experimental group was significantly more than that for the control group. Insignificant decreases in sodium consumption were identified within groups (*p* > 0.05), and the decreases between the two groups were similar (*p* > 0.05). In addition, the mean difference of sodium intake for the experimental group was more than that for the control group, although the difference was not significant. Significant increases in dietary fiber consumption were found within groups, (*p* < 0.001), and the increase in the experimental group was significantly higher than the increase in the control group (*p* < 0.001). A decrease in vitamin C intake was identified in the control group, while an increase of vitamin C intake was found in the experimental group. Moreover, the increased intake of vitamin C in the experimental group was significantly higher than that in the control group (*p* < 0.001).

### 4.3. Impact of the Dietary Monitoring App on Fruit and Vegetable Consumption

Mean daily intake of fruit and vegetables was recorded at baseline (week 1) and post-intervention (week 12). Mean (s.d.) daily fruit consumption for the control group decreased from 32.7 (36.79) grams to 26.0 (28.31) grams, while that for the experimental group increased from 37.8 (42.89) grams to 61.6 (69.98) grams. Significant differences were found within groups F_(1, 303)_ = 40.304, *p* < 0.001, but the increase in fruit intake was significantly higher in the experimental group than that in the control group (*p* < 0.001). Mean (s.d.) daily vegetable consumption for the control group increased from 233.3 (106.1) grams to 267.0 (109.0) grams, while that for the experimental group increased from 327.2 (165.1) grams to 456.2 (222.5) grams. Significant differences were identified within groups F_(1, 303)_ = 28.905, *p* < 0.001, and the increase in vegetable intake was significantly higher in the experimental group than that in the control group (*p* < 0.001) (Table 4).

### 4.4. Predictors of Fruit and Vegetable Consumption

Stepwise regression was computed with the factors of group allocation, post total GNKQ score, gender, age, and BMI categories to explore the significant factors contributing to the variances of fruit and vegetable consumption. The results showed that the intervention group was the most correlated with fruit consumption at step 1, F_(1,303)_ = 34.088, *p* < 0.001) with *r* = 0.318. The regression coefficient of step 1 was B = 35.564 and R Squared = 0.101. At step 2, BMI category exhibited the highest significant partial correlation to fruit consumption, F_(2, 302)_ = 19.488, *p* < 0.001) with *r* = 0.338. The regression coefficient of step 2 was B = 35.905 and R Squared = 0.114. No other excluded variables were determined to be significantly correlated with fruit consumption in step 2, and the stepwise regression ended at step 2.

Stepwise regression revealed that the intervention group was the most correlated with vegetable consumption at step 1, F_(1, 303)_ = 89.446, *p* < 0.001) with *r* = 0.477. The regression coefficient of step 1 was B = 189.188 and R Squared = 0.228. No other excluded variables were identified to be significantly correlated with vegetable consumption in step 1, and the stepwise regression ended at step 1.

### 4.5. Feedback on the Application of the Dietary Monitoring App

Three participating younger adults were invited to take part in an interview to collect their feedback on their experience in using the dietary monitoring app.
(a)User experience in the dietary recording process

User A recalled that she was not familiar with the dietary monitoring process, but that she became able to monitor her daily diet not long afterwards. User B needed some time to make dietary recording as a habit. She used to forget to input intake of food items, enter the daily diet records, or take photographs of new food items. However, after some practice, User B stated that she became accustomed to keeping her dietary records on a daily basis. Regarding the user interface, User A, User B, and User C found that the app was easy to use. Nevertheless, certain technical problems were found, for example, User A occasionally could not upload food photographs, and she waited for a long time for the screen to be updated. User C remarked that the dietary recording process was smooth, but that if she ate food at a higher number of portions, she needed to scroll down further to get her input correctly. She also felt that the choices of portion size with a half portion apart were not realistic for actual intake. User B also stated that the optional questions in dietary input were confusing. For instance, she was unsure if she should record sauce as consumption of soup, or if it was necessary to input the weight of food if it was not a pre-packaged food.
(b)User reflection on the whole intervention process

User A agreed that the dietary monitoring app helped her to reflect on her daily food intake, which increased her awareness of healthy eating, leading her to gradually establish good eating habits. She appreciated that the online food report and nutrient report allowed her to visualize the calorie intake and nutrient intake of each meal. For foods other than pre-packaged foods, the dietary reports facilitated User A to understand the nutrient content of her food choices. She stated that the dietary monitoring app made her reflect on three important points about healthy eating: first, she should not eat too much in a single meal; second, the intake limits of calories were determined by physical activity level; and third, food intake should be distributed over several meals. User B remarked that she was not previously aware of her nutrient intake, and that during the whole intervention process, she paid more attention to calorie intake, vitamin intake, and calcium intake in each meal. User B agreed that the whole intervention process was useful because she knew that there were nutrient labels in pre-packaged food and the nutrient information was easily found in pre-packaged food. User B stated that he had previously ignored nutrient values of food from different meals in her daily life, but that they contributed to more daily food consumption. Therefore, User B found the entire intervention process beneficial to her understanding of healthy eating. User C also remarked that the online food report and online nutrient report revealed important, and often obscured, nutritional truth. In other words, even if a food appears to be healthy, their consumption should still not exceed the recommended level, such as in the case of white rice. User C reflected that she understood that the calorie limit could be related to physical activities when she reviewed his online reports during the intervention process. In addition, before she used the dietary recording system, she thought that a high sugar level was limited to those foods that tasted sweet. After the whole intervention process, however, she knew how to avoid high sugar content foods or to choose foods with a lower glycemic index.
(c)User suggestions after the intervention

User A suggested that dietary monitoring apps should be promoted to the elderly population living in the community. NGOs could also assist the elderly to monitor their dietary intake regularly, and health educators could serve as facilitators to explain the online food reports and nutrient reports to the elderly. Overall, she believed that the elderly could especially benefit from dietary monitoring apps.

## 5. Discussion

Although the younger generation has begun to assume responsibility for their dietary habits, most of them have unhealthy eating behaviors with a low intake of fruits, vegetables, and whole grains [36,37]. Younger adults are also comfortable with technology and social media. Since their priorities in obtaining knowledge tend towards efficiency and convenience, they found this new concept of healthy eating promotion to be an appropriate way to acquire the skills to make good food choices and establish a balanced diet. Indeed, this modality of learning presents no constraints in terms of scheduled time, transportation, and location. It is also expected that this project could probably be effectively applied to other age groups to assimilate acquired nutrition knowledge with an eating plan. The process of behavioral feedback also encourages users to identify and solve problems on their own [25]. Through repeated reflective learning via behavioral feedback, other users in the community may find it easy to connect eating problems with food choices, and to implement practical dietary advice that follows recommended guidelines of healthy eating.

Prior studies suggested that nutrition knowledge is associated with higher intake of vegetables [36], and thus public health initiatives that focus on nutritional knowledge could be one sustainable option to increase healthy eating [37,38,39]. The current study indicated that the younger adults attending the nutrition seminar only, and those attending the nutrition seminar and using the dietary monitoring app, both improved their nutrition knowledge. However, having the dietary monitoring app for the purpose of dietary reflection made the younger adults more capable of matching food products to food categories, choosing healthy foods, and selecting foods to reduce health problems or risk of diseases. Healthy eating knowledge promoted to younger adults should also be unique to address their specific food access and dietary inquiries. Indeed, their individual problems in dietary patterns should be channeled to them as feedback to take effective actions. Their eating problems should be easily traced in their dietary records and dietary analyses, and immediate corresponding modifications should be made in their meal plans. To facilitate such interactive learning, dietary monitoring apps with reflective feedback are able to provide effective awareness in younger adults for changing their lifestyle in terms of healthy eating. It is found that younger adults’ self-reflection is feasible with eDietary Portal. The eDietary Portal empower younger adults to take control of identifying their individual eating problems and correcting their eating problems through proper food choices.

Regarding salt consumption, which was indicated by the outcome measure of sodium intake, the findings did not demonstrate a significant decrease from using the dietary monitoring app. However, the results indicated that the experimental group reduced its sugar consumption more than the control group. Participants may find it easy to avoid simple sugar, which is easily identified by a sweet taste. For example, they could eat less sweet-tasting foods and choose beverages without a sweet taste. Salt content in food, however, could be more difficult to avoid. Foods prepared at home could be controlled by adding less salt, or soy sauce or oyster sauce instead. It is also worth noting that salt is invisible. Indeed, foods served in restaurants could not be assessed for sodium content by the name of dishes, ingredients, or photographs of foods. This is an especially concerning issue because younger adults tend to dine-out comparatively more frequently in their daily life. This may be the primary reason that participants found it easy to increase their fruit and vegetable consumption and avoid sugar intake, but not to choose less salt in their foods.

For whole grain consumption, an increase in dietary fiber was found in both groups, with the experimental group increasing their dietary fiber statistically more than the control group. Moreover, the increase of dietary fiber intake, with a significant decrease in carbohydrate consumption, indicated a shift of food choice from starchy foods to whole grain foods. An improvement in whole grain consumption was supported in both groups, and the participants using the dietary monitoring app improved their whole grain consumption to a significantly greater extent than those in the control group. This demonstrated that participants obtained more awareness in adjusting their foods to whole grain food during the dietary monitoring process. Through the online reports, participants in the experimental group identified food choices with more dietary fiber and could continue to consume relatively more of those foods. This effect is attributed to the behavioral feedback of the reflection process, in which participants identified their eating problems by reviewing the online reports [30].

In this study, participants in the experimental group were found to consume more fruits and vegetables than those in the control group. This finding probably suggests that fruits and vegetables are comparatively easy to purchase and order, and thus to consume more of them. A recent review indicated that vegetable consumption is a less researched area when compared with fruit consumption [40]. This is attributable to the fact that research is commonly designed with fruit and vegetable consumption as an integrated outcome measurement; if separately taken into account; however, fruit consumption is usually higher than vegetable consumption [40]. The findings in the current study can provide an effective pathway to promote vegetable consumption in the process of self-dietary monitoring with behavioral feedback.

Abdel-Megeid and colleagues [41] reported that a lower BMI is associated with a higher intake of vegetables. In the present study, BMI categories improved prediction by 1.3%, indicating that higher body sizes further improved fruit consumption with dietary monitoring. However, a similar finding was not present concerning vegetable consumption. This observation was in accordance with Abdel-Megeid et al.’s [41] findings and supported the assertion that vegetable consumption is comparatively more difficult to promote among younger adults. Our results regarding BMI categories may explain that a higher BMI category influences fruit consumption, but a lower BMI category influences vegetable consumption. Further studies are required to confirm these findings.

## 6. Limitations

The present study possesses several limitations. For example, the sampling strategy was performed according to convenience, which may limit the generalizability of the findings. In addition, this project used a mobile app as the means to promote healthy eating. Consequently, participants who were more familiar with mobile technology and more accustomed to using apps may benefit more from the intervention. On the other hand, participants who may have a lower education background, lower technology literacy, less experience in using mobile apps, lower income, and work more hours per week may encounter difficulties in using this intervention for dietary reflection. However, the project team believes that, with the greater coverage of smart phone applications in the population and the well-equipped infrastructure of 4G and 5G technology in the near future, the overall population will become increasingly familiar with mobile apps. Regarding the major benefits of saving transportation time and recording dietary records entirely according to one’s own schedule, mobile apps in healthy diet promotion could be effectively generalized to other age groups, such as patients with chronic diseases, as well as elderly and minority groups, which may have relatively less access to dietary knowledge. In addition, health care providers can use these dietary monitoring apps in a highly cost-effective manner. The heathy eating tips implemented in this research contributed greatly to its success in good compliance with the dietary monitoring process. With no systematic encouragement in sending healthy eating tips, the significant results achieved by younger adults during the dietary monitoring process could be discounted. As the implementation of the dietary recording process required health educators to work along with the participants, the applications of eDietary Portal as dietary monitoring may limit scalability to a larger population to be conducted at the same time.

## 7. Conclusions

Dietary monitoring with behavioral feedback promoted younger adults to reduce sugar consumption and increase fruit and vegetable consumption. For those younger adults with increased body size, they were more likely to increase fruit consumption with the application of dietary monitoring. The effectiveness could be predicted to be markedly expanded to other populations due to increasing adoption of technology and expanding technological infrastructure.

## Figures and Tables

**Table 1 nutrients-13-00333-t001:** Demographic characteristics of the participants.

	Control	Experiment	
	*n* (%)	*n* (%)	
Sex			
Male	51 (33.1)	55 (36.4)	>0.05
Female	103 (66.9)	96 (63.6)	
Marital status			
Single	128 (83.1)	122 (80.8)	
Married	26 (16.9%)	29 (19.2)	>0.05
Number of children			
0	138 (89.6)	135 (89.4)	<0.05
1	11 (7.1)	4 (2.6)	
2	5 (3.2)	12 (7.9)	
Highest level of education			
Secondary school	19 (12.3)	11 (7.3)	<0.05
Technical or trade certificate	50 (32.5)	35 (23.2)	
Diploma	49 (31.8)	51 (33.8)	
Bachelor degree	36 (23.4)	54 (35.8)	
Living region			
Hong Kong Island	17 (11.0)	10 (6.6)	>0.05
Kowloon	41 (26.6)	33 (21.9)	
New Territories	96 (62.3)	108 (71.5)	

**Table 2 nutrients-13-00333-t002:** Chi-square tests by group assignment.

		BMI <19	19< BMI < 23	23 < BMI < 25	BMI > 25	*p*-Value
		*n* (%)	*n* (%)	*n* (%)	*n* (%)	
Male	Control group	7 (70.0)	25 (41.0)	7 (43.8)	12 (63.2)	>0.05
	Experimental group	3 (30.0)	36 (59.0)	9 (56.3)	7 (36.8)	
Female	Control group	34 (66.7)	51 (46.4)	7 (38.9)	13 (54.2)	>0.05
	Experimental group	17 (33.3)	59 (53.6)	11 (61.1)	11 (45.8)	

**Table 3 nutrients-13-00333-t003:** ANCOVA comparisons of nutrition knowledge scores.

	Control		Experiment			
	Baseline	Post	Baseline	Post	between Groups *f*-Value ^†^	*p*-Value
	Mean (s.d.)	Mean (s.d.)	Mean (s.d.)	Mean (s.d.)		
GNKQ-R ^δ^ Section 1	12.4 (2.68)	13.7 (2.80)	13.1 (2.48)	14.4 (3.54)	1.125	>0.05
GNKQ-R ^δ^ Section 2	23.6 (4.51)	25.1 (5.21)	24.6 (4.07)	27.1 (5.2)	8.264	<0.05
GNKQ-R ^δ^ Section 3	8.1 (2.42)	8.5 (2.39)	8.5 (2.34)	9.3 (2.76)	4.931	<0.05
GNKQ-R ^δ^ Section 4	14.3 (2.88)	14.6 (2.75)	14.0 (2.69)	15.1 (3.61)	4.050	<0.05
GNKQ-R (Total Score)	58.4 (9.61)	61.9 (10.57)	60.1 (8.67)	65.9 (13.25)	5.948	<0.05

^†^ ANCOVA analysis on post-GNKQ-R scores as the dependent variable and pre-GNKQ-R scores as the controlled independent variables. ^δ^ Section 1 = knowledge about healthy food recommendations; Section 2 = knowledge about food group classification; Section 3 = knowledge about choosing healthy foods; Section 4 = knowledge about diet-related health problems or diseases.

**Table 4 nutrients-13-00333-t004:** Comparisons of nutrient intake.

	Control		Experiment			
	Baseline	Post	Baseline	Post	between Groups f-Value ^†^	*p*-Value
Energy requirement (calories)	2182 (332)	2197 (322)	2386 (433)	2420 (447)	5.414	*p* < 0.05
Energy intake (calories)	2278 (474)	2036 (395)	2551 (827)	2045 (613)	35.317	*p* < 0.001
Carbohydrate intake (grams)	231 (69)	215 (154)	244 (86)	205 (59)	23.503	*p* > 0.001
Protein intake (grams)	166 (39)	149 (46)	158 (62)	129 (53)	15.274	*p* < 0.001
Total fat intake (grams)	76.7 (32.7)	64.2 (22.6)	104.7 (49.8)	78.5 (38.3)	3.255	*p* > 0.05
Saturated fat intake (grams)	54.7 (24.3)	48.6 (17.5)	75.3 (37.0)	57.3 (29.4)	14.397	*p* < 0.001
Trans fatty acid intake (grams)	2.4 (0.97)	2.2 (0.88)	2.7 (1.32)	1.33 (1.08)	112.390	*p* < 0.05
Cholesterol intake (milligrams)	446 (163)	480 (207)	514 (239)	492 (411)	0.577	*p* > 0.05
Sugar intake (grams)	72.4 (48.3)	60.2 (34.4)	100.4 (55.8)	53.8 (36.9)	53.645	*p* < 0.001
Dietary fibre intake (grams)	10.3 (4.67)	11.8 (4.78)	14.4 (7.26)	21.2 (11.56)	48.625	*p* < 0.001
Sodium intake (milligrams)	3611 (1164)	3794 (3653)	3590 (1498)	3659 (3989)	0.082	*p* > 0.05
Calcium intake (milligrams)	299 (140)	312 (151)	328 (189)	594 (420)	60.109	*p* < 0.001
Vitamin C intake (milligrams)	25.8 (24.4)	22.9 (20.4)	35.4 (34.7)	55.4 (46.7)	64.442	*p* < 0.001
Vegetable consumption (grams)	233 (106)	267 (109)	327 (165)	456 (223)	49.555	*p* < 0.001
Fruit consumption (grams)	32.7 (36.8)	26.1 (28.3)	37.8 (42.89)	61.6 (70.0)	43.258	*p* < 0.001

^†^ ANCOVA analysis on post-energy requirement, post-energy intake, and post-nutrient intake as dependent variables, and pre-energy requirement, pre-energy intake, and pre-nutrient intake as the controlled independent variables, respectively.

## Data Availability

Data available on request due to restrictions in external funding support.

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
