# Peer review of "Younger Adults Are More Likely to Increase Fruit and Vegetable Consumption and Decrease Sugar Intake with the Application of Dietary Monitoring"

_nutrients, 2021, doi:10.3390/nu13020333_

Round 1
Reviewer 1 Report
In this study Chung et al evaluated the effect of diet monitoring applications on nutrition knowledge and eating habits in a group of young adults. The authors found that the use of the mobile apps, promoting nutrition knowledge and healthy eating, was associated with a reduced consumption of fat and sugar and with an increased intake of fibre, calcium, Vitamin C, vegetable and fruit.
The subject of the study is of particular interest, since promoting a healthy lifestyle through updated tools must be a priority for all Countries.
However, some minor issues should be addressed:
- the description of the results is excessively technical and schematic, this makes it difficult to follow and to understand. I suggest a simpler description supported by the rationale of the comparison and the meaning of the results observed.
- Information regarding apps users satisfaction are missing
- In the limit section it could be interesting to comment the actual use of these apps in a larger population outside the trial. According to authors, will people use them even if not systematically encouraged to do so?
Author Response
Responses to Reviewer 1
In this study Chung et al evaluated the effect of diet monitoring applications on nutrition knowledge and eating habits in a group of young adults. The authors found that the use of the mobile apps, promoting nutrition knowledge and healthy eating, was associated with a reduced consumption of fat and sugar and with an increased intake of fibre, calcium, Vitamin C, vegetable and fruit.
The subject of the study is of particular interest, since promoting a healthy lifestyle through updated tools must be a priority for all Countries.
However, some minor issues should be addressed:
- the description of the results is excessively technical and schematic, this makes it difficult to follow and to understand. I suggest a simpler description supported by the rationale of the comparison and the meaning of the results observed.
Responses: Thanks for your comments. Section 4.1, Section 4.2, Section 4.3 and Section 4.4 had been revised to a more readable style to describe the results directly. Between groups F values were inserted to Table 3 and Table 4 instead of in-text, making the results easier for reference.
- Information regarding apps users satisfaction are missing
Responses: The research team did not conduct apps user satisfaction quanititatively. However, the research team conducted feedback interviews with three participants. The feedback were verbatium transcripted. In the revised manuscript, the feedback on their experience in dietary monitoring process, reflection on the whole intervention process and user suggestion. Their feedback had been summarized and inserted to Section 4.5.
- In the limit section it could be interesting to comment the actual use of these apps in a larger population outside the trial. According to authors, will people use them even if not systematically encouraged to do so?
Responses: Thanks for your suggestion. I agreed that if systematic encouragement was not adopted, the results could be less promising. In the revised manuscript, I added this part to the “Limitation” session, “Regarding the compliance of dietary monitoring process, the systematic encouragement implemented in this research paid valuable effort to its success. With systematic encouragement, the significant results achieved by younger adults during the dietary monitoring process could be discounted.”
Reviewer 2 Report
This is a very interesting paper except section 4.1 was so technically written that it was very difficult for me to clearly understand what the results conveyed. Also, please double check line 38 as I think the 20 times is more like 10 times from my review of the paper.
Author Response
Responses to Reviewer 2
This is a very interesting paper except section 4.1 was so technically written that it was very difficult for me to clearly understand what the results conveyed.
Responses: Thanks for your comments. Section 4.1, Section 4.2, Section 4.3 and Section 4.4 had been revised to a more readable style to describe the results directly. Between groups F values were inserted to Table 3 and Table 4 instead of in-text, making the results easier for reference.
Also, please double check line 38 as I think the 20 times is more like 10 times from my review of the paper.
Responses: For line 38, the risk of progressive metabolic syndrome with higher BMI was cited from reference 10 (Lloyd-Jones et al, 2007). I made the judgement from Table 2 in the publication and I copied my calculation here for your information.
|
Sex |
Men |
Women |
||||||
|
Risk factor |
Stable BMI |
Flucutating BMI |
Increased BMI |
Ratio |
Stable BMI |
Flucutating BMI |
Increased BMI |
Ratio |
|
Baseline BMI 20.0-24.9 kg/m2 |
|
|
|
|
|
|
|
|
|
Trigycerides |
1.01 |
1.00 |
4.34 |
4.3 |
0.25 |
0.07 |
1.84 |
7.4 |
|
HDL cholesterl |
0.04 |
-0.24 |
0.54 |
13.5 |
0.18 |
0.34 |
0.25 |
1.4 |
|
Fasting glucose |
0.08 |
-0.01 |
0.31 |
3.9 |
0.06 |
-0.09 |
0.27 |
4.5 |
|
Fasting insulin |
0.07 |
0.00 |
0.27 |
3.9 |
0.08 |
-0.1 |
0.20 |
2.5 |
|
Systolic BP |
0.14 |
-0.04 |
0.31 |
2.2 |
0.15 |
0.39 |
0.65 |
4.3 |
|
Diastolic BP |
0.41 |
0.36 |
0.57 |
1.4 |
0.31 |
0.44 |
0.64 |
2.1 |
|
Baseline BMI 25.0-29.9 kg/m2 |
|
|
|
|
|
|
|
|
|
Trigycerides |
0.84 |
1.65 |
4.11 |
4.9 |
0.82 |
0.69 |
2.27 |
2.8 |
|
HDL cholesterl |
0.18 |
-0.16 |
0.46 |
2.6 |
0.29 |
0.14 |
0.38 |
1.3 |
|
Fasting glucose |
0.04 |
0.84 |
0.34 |
8.5 |
0.12 |
0.6 |
0.46 |
3.8 |
|
Fasting insulin |
0.02 |
-0.04 |
0.42 |
21.0 |
0.32 |
-0.17 |
0.376 |
1.2 |
|
Systolic BP |
0.17 |
0.36 |
0.38 |
2.2 |
0.03 |
0.35 |
0.83 |
27.7 |
|
Diastolic BP |
0.32 |
0.45 |
0.61 |
1.9 |
0.11 |
0.29 |
0.68 |
6.2 |
I further revised the lines to reflect clearly with my interpretation on the cited information as “This trend is alarming because higher body mass index in early adulthood increases the risk of progressive metabolic syndrome by 21 times the most for men and 27 times the most for women in the next 15 years [10].”
Reviewer 3 Report
Thank you for an interesting paper.
I have provided comments below by line numbers. I do think the paper needs a strong English language edit. The phrasings and choice of words in many instances needs to be improved. Last - the name of the paper is mismatched with the results/discussion ie it does not feature at all in the discussion.
line 30: check these statistics. 35% and 38.9% obese seems very high for this age group. Likely overweight or obese.
line 32 overweight or obese
Line 32 - what is young aged group?
line 48 - ref needed to support this statement
line 57-58 - ref needed
line 66 - is 39 considered younger adults? I would expect this age group to be up to 24 or 29 perhaps. Any literature that classifies up to 39 as young adult?
line 67 - you indicate apps. However, I think you have tested/evaluated just a single app. If so - need to change this throughout. Plus - when first using app - you need to use full word - application - and then indicate from now on will use the shortened term of app.
line 73 - wouldn't this be decrease the consumption of salts and sugars?
line 82 - is not mediating the transfer but facilitating the application
line 82-84 - starting with Using the apps..This is not about method. remove
lines 78-94: this needs to be reworked. The design is quasi-experimental 2 group design. The sampling is convenience sampling at the 12 selected locations with alternate assignment to the two groups. Need to explain what are the NGOs.
line 101 - you indicate the nutrition information sessions were conducted in class. Does this mean all participants were university students? But, you indicate recruitment was at 10 NGOs.
Line 105-113: this needs to be better explained. it is difficult to understand what was actually delivered and what the participants had to do.
line 120 - remove adminstered
ine116-125 - add some information on the scale's psychometric properties.
line136-139 - this indicates the sample self-selected. This is a serious limitation to the study. Needs to be discussed.
line150-151 this is a statement of effect. Not appropriate in the methods section
line 153 - the word learned is an effect. need to use a neutral term here.
line 154-156 - again, this is a statement of effect. It is a methods section - just the facts no embellishment or statements indicating effect or value are appropriate.
line 185-186 - you indicate randomisation stratification but earlier you have reported group allocation was done based on alternate groups. You have now described 3 different group allocation processes!
line 203-207. OK there was some difference but you have not indicated is the difference significant between control and intervention groups after accounting for the pre/post differences in both groups. We can see from the table that the control group also improved (and that there were differences between the two groups at baseline)- so - need to partition this out as noise to determine if there is a real difference between control and intervention groups that may be linked to the app.
line 277-278 - what does this mean?
Line325-337 - need to discuss sampling strategy and limitations to generalisability of findings. Also, perhaps the use of researchers to analyse photos and how this could introduce subjectivity. Also - make the intervention not scalable to a larger population - this is a major limitation
Author Response
Responses to Reviewer 3
Thank you for an interesting paper.
I have provided comments below by line numbers. I do think the paper needs a strong English language edit. The phrasings and choice of words in many instances needs to be improved.
Responses: A strong English language editing expert was requested for phrasings and choice of words improvement for the whole manuscript.
Last - the name of the paper is mismatched with the results/discussion i.e it does not feature at all in the discussion.
Responses:
line 30: check these statistics. 35% and 38.9% obese seems very high for this age group. Likely overweight or obese.
Responses: After checking with the statistics, it should be 35% of US young adults and 27.9% of the Australian young population are found obese.
line 32 overweight or obese
Responses: For the “50.0% Hong Kong adults aged 15-84, the statistics referred to overweight and obese, collectively.
Line 32 - what is young aged group?
Responses: The young aged group are under 30. I have inserted this back to the manuscript.
line 48 - ref needed to support this statement
Responses: Agreed with your observation. I have inserted two references to support this statement.
They are:
Poobalan, A.S., Aucott, L.S., Clarke, A., Smith, W.C.S. Diet behavior among young people in transition to adulthood (18-25 year olds): a mixed method study. Health Psycho9l Behav Med 2014, 2(1), 909-928.
Munt, A.E., Partridge, S.R., Allman-Farinelli, M. The barriers and enablers of healthy eating among young adults: a missing piece of the obesity puzzle: A scoping review. Obes Rev 2017, 18(1), 1-17.
line 57-58 - ref needed
Responses: Agreed with your observation. I have inserted two references to support this statement.
They are:
Duncan, M.J., Vandelanotte, C., Rosenkrantz, R.R., et al. Effectiveness of a website and mobile phone based physical activity and nutrition intervention for middle-aged males: trial protocol and baseline findings of the ManUp Study. BMC Pub Health, 2012, 12, 656.
Chung, L.M.Y., Fong, S.S.M. Role of Behavioural Feedback in Nutrition Education for Enhancing Nutrition Knowledge and Improving Nutritional Behaviour among Adolescents. Asia Pac J Clin Nutr 2018, 27(2), 466-472.
line 66 - is 39 considered younger adults? I would expect this age group to be up to 24 or 29 perhaps. Any literature that classifies up to 39 as young adult?
Responses: According to the position statement of the Society for Adolescent Health and Medicine (2017), there are no clear guidelines for determining what ages should be included in the designation of young adulthood. Young people in some cases could be extended to 30 or 40 years. As the project would like to include more eligible participants to the study, we adopted the extended scope of designations in the society.
line 67 - you indicate apps. However, I think you have tested/evaluated just a single app. If so - need to change this throughout. Plus - when first using app - you need to use full word - application - and then indicate from now on will use the shortened term of app.
Responses: Thanks for your comments. I have revised the term in the manuscript to make it more clear to its representation. Here is the revised writing: “There is also few intervention applying mobile applications (the apps) to improve younger adults’ nutrition behaviour. In this study, a dietary monitoring mobile apps called “eDietary Portal” integrated with the component of behavioural feedback was evaluated in the younger adults aged 19-39.”
line 73 - wouldn't this be decrease the consumption of salts and sugars?
Responses: Agreed and I changed it to “decrease”.
line 82 - is not mediating the transfer but facilitating the application
Responses: Agreed and I changed it to “facilitating”.
line 82-84 - starting with Using the apps..This is not about method. Remove
Responses: Agreed and I removed the sentences “Using the apps as experiential learning and reflective learning, it was expected the participants acquired the skill in healthy food choices and changed their eating behaviour accordingly” from the manuscript.
lines 78-94: this needs to be reworked. The design is quasi-experimental 2 group design. The sampling is convenience sampling at the 12 selected locations with alternate assignment to the two groups. Need to explain what are the NGOs.
Responses: Thanks for your comments and suggestions. The Design and Setting was revised with the suggested amendment. About the NGOs, they were invited to recruit participant and to provide venue for the research team to conduct nutrition education. The NGOs were those providing community services to population living in the districts. I have inserted these elaboration to the manuscript.
line 101 - you indicate the nutrition information sessions were conducted in class. Does this mean all participants were university students? But, you indicate recruitment was at 10 NGOs.
Responses: The nutrition information sessions were conducted in group for effective arrangement. There were university students but there were working groups too. Participants if they were university students, they were recruited by the 2 institutions. Participants if they were working groups, they were recruited by the 10 NGOs. The group size ranging from 12 – 50 and in the nearest centre/lecture room the participants found convenient to them.Line 105-113: this needs to be better explained. it is difficult to understand what was actually delivered and what the participants had to do.
Responses: This part had been drastically amended as below:
Physical measurements of the participants and their age, sex and daily physical activity levels were input to the apps for both control group and experimental group. Participants in the experimental group recorded their daily intake on the dietary monitoring apps by taking photos of their foods and inputting the intake serving. Dietary reports with nutrient analysis were shown online for the participants to review. The participants reflected their dietary intake by reviewing the online reports to check if there were excessive intake in undesirable nutrients or deficiency intake in desirable nutrients. They were asked to change their food choices day by day and ate towards their healthy goal. Their self-reflection and change in eating patterns were recorded by the apps in the central database. The nutritional knowledge the participants received in the nutrition education and reflection in daily recording and reviewing process facilitated the participants to acquire healthy eating. A nutrition behaviour established in the process of reflective learning generated by the reflective feedback was expected [30]. Participants in the control group attended the 3 hours nutrition education with the content similar to those in the experimental group. The participants in the control group only recorded their diet in the first week and the twelfth week as baseline and post intervention measurements. They did not monitor their dietary pattern between the pre-post measurements. No online report with behavioural feedback was involved.
line 120 - remove administered
Responses: This word had been removed from the sentence.
ine116-125 - add some information on the scale's psychometric properties.
Responses: Psychometric properties of GNKQ-R had been inserted to the manuscript. They are: GNKQ-R had high internal reliability with Cronbach’s alpha = 0.93 and that of each section ranged from 0.70 to 0.86. The intraclass correlation coefficients for the external reliability ranged 0.72-0.89. The construct validity was tested with effect size ranged from 0.5 to 1.2.line136-139 - this indicates the sample self-selected. This is a serious limitation to the study. Needs to be discussed.
Responses: I did not understand your comments about “the sample self-selected”. Did you refer to the section “3.4.2”? Indeed, the dietary behavior was measured by the nutrient values analyzed from the dietary records.
line150-151 this is a statement of effect. Not appropriate in the methods section
Responses: Those statements related to the effect had been removed from the methods section.
line 153 - the word learned is an effect. need to use a neutral term here.
Responses: Those statements related to the effect had been removed and the methods section amended should be more neutral and descriptive.
line 154-156 - again, this is a statement of effect. It is a methods section - just the facts no embellishment or statements indicating effect or value are appropriate.
Responses: Those statements related to the effect had been removed and the methods section amended should be more neutral and descriptive.
line 185-186 - you indicate randomisation stratification but earlier you have reported group allocation was done based on alternate groups. You have now described 3 different group allocation processes!
Responses: Sorry for the confusing terms misused in the manuscript. The revision should look more consistent in convenient group assignment between two groups only.
line 203-207. OK there was some difference but you have not indicated is the difference significant between control and intervention groups after accounting for the pre/post differences in both groups. We can see from the table that the control group also improved (and that there were differences between the two groups at baseline)- so - need to partition this out as noise to determine if there is a real difference between control and intervention groups that may be linked to the app.
Responses: The results session had been drastically amended to make it easier for comparison. I also insert the between groups F values to Table 3 and Table 4 instead of in-text, making the results easier for group comparison to highlight the significant difference (if any) between control and intervention groups that may be linked to the apps.
line 277-278 - what does this mean?
Responses: I have rephrased this part. It means BMI followed by intervention, may possibly relating to the variance of fruit consumption, but not vegetable consumption.
Line325-337 - need to discuss sampling strategy and limitations to generalisability of findings.
Responses: I have added this statement to the limitation, “As the sampling strategy was by convenience which may limit the generalizability of the findings.”
Also, perhaps the use of researchers to analyse photos and how this could introduce subjectivity.
Responses: The food photos provided the serving sizes of food portion. This in fact was more accurate than the self-reported written conventional 24 hour dietary recall which usually deemed as gold standard. About the nutrient analysis on the food intake, the researcher did the analysis with Nutritionist Pro which is a widely adopted database with over 82,000 foods with nutrient details from the brands directly and from various government sources. Therefore, the process of food analysis attached to the common practice in research.
Also - make the intervention not scalable to a larger population - this is a major limitation.
Responses: I have added this statement to the limitation, As the implementation of dietary recording process required health educators to work along with the participants, the applications may limit the scalability to a larger population to be conducted at the same time.
Round 2
Reviewer 3 Report
Thank you for this revised paper. English language is much improved.
A few minor issues and a couple of results issues that need to be addressed.
Lines 80-87: need to fix numbering
Line 185: grammar issues
Line 185-186: you indicate this strategy worked to motivate them. But, this is the methodology section. This needs to be neutral.
Line 227: significantly
Line 235-236: this analysis needs to be checked. For GNKQ-R ⸹Section 1 while the two groups have different pre and post scores - the change over time is the same at 1.3. Therefore no difference between the groups. Therefore, the difference between the groups for this section of GNKQ-R cannot be significantly different as stated.
Line 283-284: this needs work. What factors were included in the stepwise regression. What is the rationale for including the groups as factor - seems unusual to me.
line 288 & 292: do you mean included? rather than excluded.
line 339: replace his with her
line 354: replace a remarkable with an appropriate
line 355: replace correct with good
line 360: ref needed at end of sentence ending with own
Line 365: remove ed - focus
line 424: change Abdel-Megeid et al. [41]’s to Abdel-Megeid et al.'s [41]
The title does not reflect the paper. The finding of younger adults with increased body size is not a central finding - certainly not as the paper is currently presented. It is not discussed. The title should be changed to better reflect the paper content.
Author Response
Responses to Reviewer 3
A few minor issues and a couple of results issues that need to be addressed.
Response: Thanks for your patience in review and giving me further comments. The revision of the manuscript had been made with the following details.
Lines 80-87: need to fix numbering
Response: The numbering of the 4 research questions had been fixed with a second level.
Line 185: grammar issues
Response: The original sentence of line 185 was “The nutrient analysis of the food items was carried out by the researchers. “ which seems nothing wrong with the grammar. Could you let me know the issue in specific?
Line 185-186: you indicate this strategy worked to motivate them. But, this is the methodology section. This needs to be neutral.
Response: I replaced the word “motivate” by “enhance” to revise the sentence to be neutral.
Line 227: significantly
Response: Amended.
Line 235-236: this analysis needs to be checked. For GNKQ-R ⸹Section 1 while the two groups have different pre and post scores - the change over time is the same at 1.3. Therefore no difference between the groups. Therefore, the difference between the groups for this section of GNKQ-R cannot be significantly different as stated.
Response: Thanks for your observation. The data analysis had been checked and amended accordingly.
Line 283-284: this needs work. What factors were included in the stepwise regression. What is the rationale for including the groups as factor - seems unusual to me.
Response: Stepwise regression was computed with the factors of group allocation, post total GNKQ score, gender, age and BMI categories to explore the significant factors contributing to the variances of fruit and vegetable consumption. These elaboration had been inserted to the line 276-278. The rationale for including the groups as factor was to explore its contribution to the variance of dependent variables as compared with other independent variables.
line 288 & 292: do you mean included? rather than excluded.
Response: In stepwise regression, only significant partial correlation of independent variables for predicting dependent variables were included. Those independent variables were excluded in the model if their partial correlations were not significant. In the sentences, the meanings were excluded.
line 339: replace his with her
Response: Thanks for your observation. This was replaced from his with her.
line 354: replace a remarkable with an appropriate
Response: Thanks for your suggestion. It had been replaced.
line 355: replace correct with good
Response: Thanks for your suggestion. It had been replaced.
line 360: ref needed at end of sentence ending with own
Response: Reference had been inserted at the end of the sentence.
Line 365: remove ed – focus
Response: The ed of focus had been removed.
line 424: change Abdel-Megeid et al. [41]’s to Abdel-Megeid et al.'s [41]
Response: The format of in-text reference had been changed to Abdel-Megeid et al.'s [41].
The title does not reflect the paper. The finding of younger adults with increased body size is not a central finding - certainly not as the paper is currently presented. It is not discussed. The title should be changed to better reflect the paper content.
Response: Thanks for your recommendation. I had amended the title to “Younger adults are more likely to increase fruit and vegetable consumption and decrease sugar intake with the application of dietary monitoring.” Which would be more suitable and better reflect the paper content.